# The Creep-Sliding Deformation Mechanism of the Jiaju Ancient Landslide in the Upstream of Dadu River, Tibetan Plateau, China

Yiqiu Yan [1], Changbao Guo [1,2,*], Caihong Li [1,3], Hao Yuan [1] and Zhendong Qiu [1]

1 Institute of Geomechanics, Chinese Academy of Geological Sciences, Beijing 100081, China
2 Key Laboratory of Active Tectonics and Geological Safety, Ministry of Natural Resources, Beijing 100081, China
3 School of Earth Sciences and Resources, China University of Geosciences, Beijing 100083, China
* Correspondence: guochangbao@cags.ac.cn

**Abstract:** The Jiaju ancient landslide is a giant landslide located upstream of the Dadu River, eastern Tibetan Plateau, with a volume of approx. $7.04 \times 10^8$ m$^3$. The Jiaju ancient landslide is complex and comprises five secondary sliding bodies, e.g., the Jiaju landslide (H01), Niexiaping landslide (H02), Xiaobawang landslide (H03), Niela landslide (H04), and Mt.-peak landslide (H05). Affected by regional neotectonic movement, heavy rainfall, river erosion, and lithology, the secondary sliding bodies of the Jiaju ancient landslide are undergoing significantly different creep-sliding deformation, which will cause great damage to villages, roads, and rivers around the sliding bodies. Combined with the SBAS-InSAR method, Sentinel-1A data from June 2018 to August 2021, remote sensing and field surveys, this study obtained the Jiaju ancient landslide deformation characteristics and deformation rate in the line-of-sight direction ($V_{LOS}$), slope ($V_{Slope}$), and vertical ($V_{Vertical}$). It is concluded that the maximum deformation rate of the Jiaju ancient landslide is significant. The maximum of $V_{LOS}$, $V_{Slope}$, and $V_{Vertical}$ are $-179$ mm/a, $-211$ mm/a, and $-67$ mm/a, respectively. The Niela landslide (H04), Jiaju landslide (H01), and Mt.-peak landslide (H05) are very large and suffer strong deformation. Among these, the Niela landslide (H04) is in the accelerative deformation stage and at the Warn warning level, and the Jiaju landslide (H01) is in the creep deformation and attention warning level, especially heavy rainfall, which will accelerate landslide deformation and trigger reactivation. Because the geological structure is very complex for the Jiaju ancient landslide and strong neotectonic movement, under heavy rainfall, the secondary landslide creep-sliding rate of the Jiaju ancient landslide is easily accelerated and finally slides in part or as a whole, resulting in river blocking. It is suggested to strengthen the landslide deformation monitoring of the Niela landslide and Jiaju landslide and provide disaster mitigation and prevention support to the government and residents along the Dadu River watershed.

**Keywords:** Danba Jiaju; ancient landslide; SBAS-InSAR; deformation monitoring; stability analysis



## 1. Introduction

Landslides, defined as the movement of a mass of rock, debris, or earth down a slope [1,2], can be triggered by a variety of external factors, such as intense rainfall, earthquakes, water level changes, and human engineering activities [3,4]. Landslides have developed throughout the world, especially in the seismic zone, active fault zone, loess region, mountain canyon region and rainstorm region, e.g., the region of British Columbia in Canada [5], the Monesi di village in Italy [6], the Baglihar reservoir in India [7], the Pakistan highway in Karakorum [8], and the Sichuan, Tibet, Gansu province in China [9–12]. The Tibetan Plateau is one of the places with the most severe landslide hazard distribution in China, and a series of large-scale landslides occurred [13–15], such as the 2000 Yigong long run-out landslide that was active in Tibet, China [16], with a volume

of $2.8 \times 10^8$~$3.0 \times 10^8$ m$^3$ [17]; the Baige giant landslide occurred twice in October and November 2018 successively and blocked the Jinsha River [18,19]. The giant landslide hazard chains have significantly harmed urban residents and engineering construction upstream and downstream of landslides. According to the geological formation age, landslides are divided into ancient landslides, old landslides, and modern landslides. Among them, the deformation and reactivation of ancient landslides seriously affect the safety of human life [15]. In addition to sudden newly occurring landslide hazards, large-scale ancient landslides and their reactivation hazards are severe in the Tibetan Plateau. For example, the Jiangdingya ancient landslide was reactivated in July 2018 in Zhouqu County, Gansu Province, and the reactivation volume reached $480 \times 10^4$~$550 \times 10^4$ m$^3$, dammed the Bailong River, and flooded the village and power station in Nanyu [20,21]. Danba County is located upstream of the Dadu River, Sichuan Province. Under the influence of tectonic activities, a series of large-giant ancient landslides developed in this area and were reactivated by deep creeping. For example, in 2004, the Jianshejie landslide was reactivated, and several large cracks or even collapsed damages appeared at the landslide's front edge and significantly threatened the houses and people in Danba County [22]. In June 2020, due to heavy rainfall, the Aniangzhai ancient landslide began to slide and induced a dammed lake, which caused severe hazards and S303 provincial road cutting off [23]. The development characteristics and their reactivation mechanisms for ancient landslides upstream of the Dadu River have become a research hotspot.

InSAR (Interferometric Synthetic Aperture Radar) is a quick and accurate landslide detection method. It can uniquely calculate subtle ground-surface movement with high accuracy [24–28]. Numerous data processing methods have been developed recently, including D-InSAR, SBAS-InSAR, DS-InSAR, and PS-InSAR. Different InSAR techniques have demonstrated their distinctive capabilities in regional landslide detection. The application of InSAR to landslide studies was first used in the 1990s, when the French scholar Achache [29] used SAR data obtained from the ERS-1 satellite to obtain ground deformation interferograms by D-InSAR technology processing. The SBAS-InSAR method provides the problem of decoherence induced by a long spatiotemporal baseline and reduces the influence of atmosphere and topography, which has been widely used in landslide monitoring [30], slope stability in northwestern Sicily of Italy [31], and Xiongba ancient landslide creeping study along the Jinsha River in China [28]. The PS-InSAR technology is usually implemented by computing differential interferograms of all the acquisitions with respect to the same reference image [32–34].

The frequent occurrence of landslides on the Tibetan Plateau and the continuous deformation of some large-scale landslides have posed significant challenges to the stability of regional geohazards, hazard prevention, and mitigation. Zheng et al. [35] and Yin et al. [36] monitored the displacement change for the Jiaju landslide by GPS and concluded that the north side displacement was more prominent than the south from August 2006 to December 2007. Bai et al. [37] studied the dynamic movement process, instability, and failure signs for Jiaju landslides, based on GPS, InSAR, and inclinometer data and found that the Jiaju landslide was in the evolution stage of accelerated movement on the surface and slow deep deformation. Dong et al. [38] combined PS-InSAR and DS-InSAR and determined that the Jiaju landslide moved faster in the north than in the south, and the maximum rate was 120 mm/a in the LOS direction. Field surveys and remote sensing interpretation found that the Jiaju ancient landslide comprises five secondary landslides, including the Jiaju landslide (H01), Niexiaping landslide (H02), Xiaobawang landslide (H03), Niela landslide (H04), and Mt.-peak landslide (H05). The Jiaju ancient landslide has the characteristics of large scale, complex formation mechanism, and obvious reactivation. In this study, the SBAS-InSAR method is used to analyze the development characteristics of the Jiaju ancient landslide and propose the main influencing factors and trends of landslide deformation. Based on the two-dimensional deformation, the line-of-sight (LOS) is converted into the slope and vertical velocity, then based on the stability trend analysis model the development stages and warning levels for the Jiaju ancient landslide are concluded.

The research results are essential for hazard mitigation and prevention of the Dadu River drainage and throughout the world.

## 2. Geological Background

### 2.1. Geomorphology

The Jiaju ancient landslide is located upstream of the Dadu River Basin, eastern Tibetan Plateau (Figures 1 and 2a), on the west side of the Dajin River [39]. The investigation area is characterized by steep topography and alpine canyons ranging from 1800 m to 5200 m. Danba County is located in the monsoon climate zone of the Tibetan Plateau, and the annual precipitation is moderate, concentrated in summer, with an average of 600 mm.

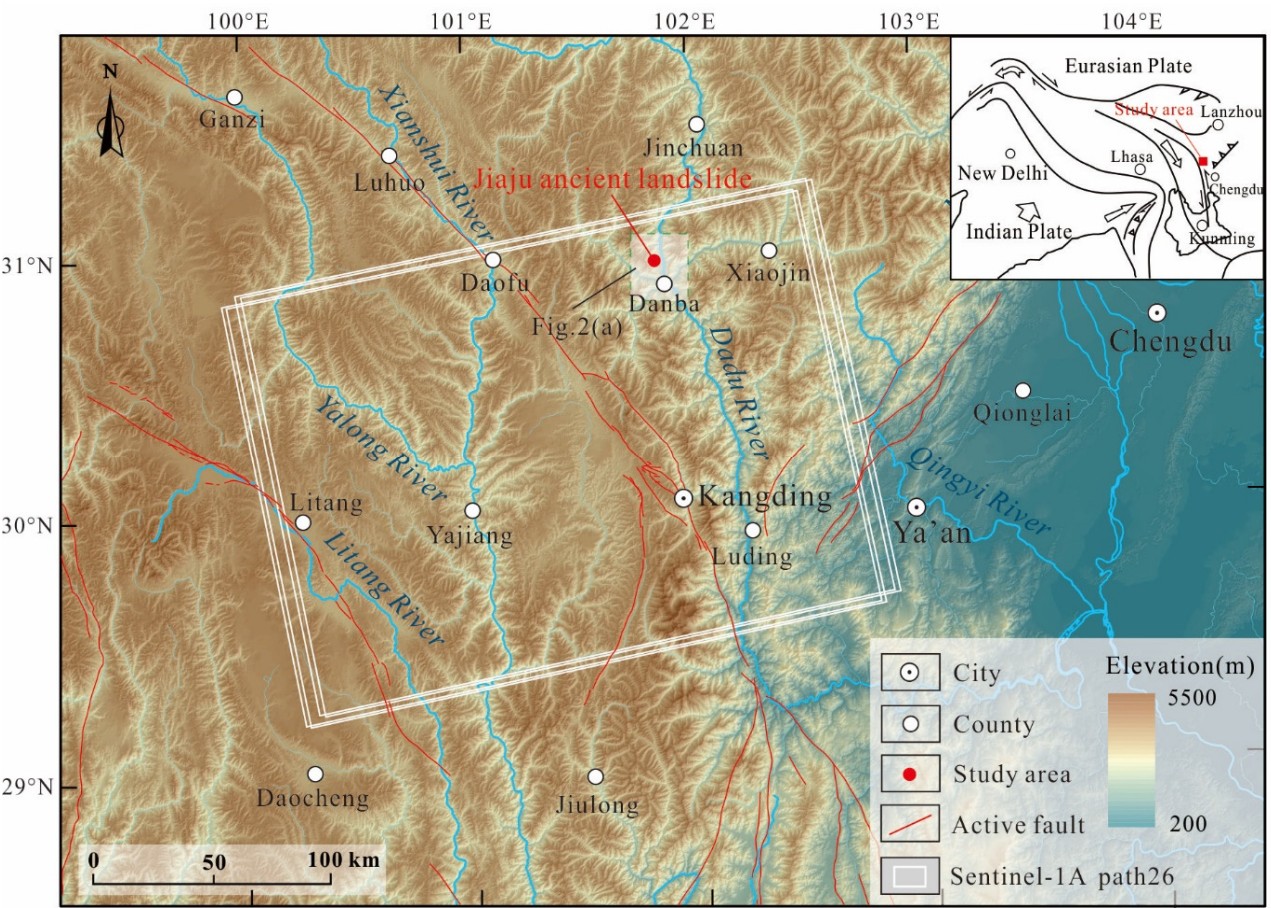

**Figure 1.** The Jiaju ancient landslide location and the Sentinel-1A ascending data coverage map.

### 2.2. Structure and Lithology

It is in the composite site of the north-south structure of Sichuan and Yunnan and the Xiaojin-Jintang arc structure. The tectonic movement in the area is complex, metamorphism and magma eruption are strong, and the structural traces are mainly northwest. Nearly north-south trending and arc-shaped fault zones are developed at the intersection of faults [40,41]. The Tanchanggou (TCG) fault and the Foyeya (FYY) fault pass through the Jiaju ancient landslide (Figure 2a,b). Affected by tectonic movement, the strata are partially inverted, with intense extrusion deformation, joint fissure development, and strong magmatic activity. Pegmatite dikes are exposed in the area, and the Silurian Maoxian Group is mainly exposed in the Jiaju ancient landslide. The schist of the Fourth Formation is sandwiched with striped marble ($Smx^4$), and the third formation of the Silurian Maoxian Group is quartzite sandwiched with schist ($Smx^3$) (Figure 2a). The strata on both sides of the landslide front are oriented from south to north. Gradual changes occur from NNW to NEW, and the inclination angle is generally 50~55° [42].

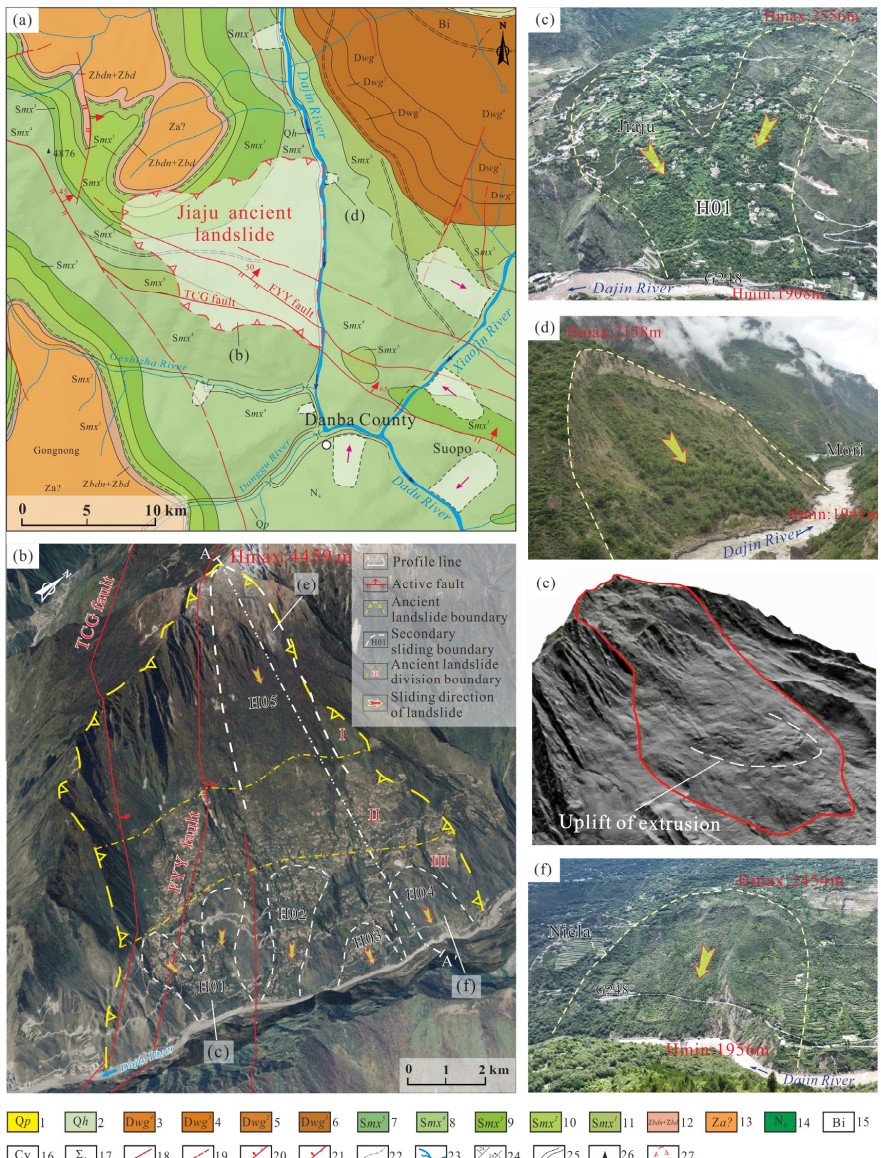

**Figure 2.** (**a**) Regional tectonic location map; (**b**) remote sensing interpretation of the Jiaju ancient landslide; (**c**) Jiaju landslide UAV image (Mirror to SW); (**d**) Genba landslide (Mirror to SE); (**e**) LiDAR image of Mt.-peak landslide (according to [43]); (**f**) Jiaju landslide UAV image (Mirror to SW). 1—Pleistocene gravel clay; 2—Holocene gravel clay; 3—Mud Quartzite, (carbonaceous) schist, and marble of the fourth rock group of the Weiguan Group of the basin system; 4—Metaconglomerate, schist, and marble of the third rock group of the Weiguan Group of the Devonian system; 5—the Weiguan group of the Devonian system schist and marble of the second rock group; 6—the first rock group of the Weiguan Group of the Devonian Quartzite; 7—the fifth rock group of the Silurian Maoxian Group quartzite and schist; 8—the fourth rock of the Silurian Maoxian Group Formation schist with banded marble; 9—the third group of Silurian Maoxian Group quartzite with schist; 10—the second group of Silurian Maoxian Group schist with marble; 11—the first of the Silurian Maoxian Group Rock formation schist; 12—Sinian Upper Dengying Formation Doushantuo Formation dolomitic marble and schist; 13—Sinian Lower Sinian granulite, conglomerate schist, migmatite; 14—Hercynian unfractionated intrusive rock; 15—biotite belt; 16—lanite belt; 17—undivided ultrabasic rock; 18—measured fault; 19—inferred fault of unknown nature; 20—measured normal fault; 21—measured reverse fault; 22—inferred geological boundary; 23—river; 24—stratigraphic inclination and dip angle, inversion of bottom layer occurrence; 25—measured geological boundary; 26—elevation point; 27—Jiaju ancient landslide.

### 2.3. Geohazards

The upstream Dadu River is located in an area with high-risk geological hazards in Southwest China, and the hazards are mainly intensively distributed along both sides of the Dadu River, such as the Dajin River and Xiaojin River. There are many newly occurring landslides or reactivated ancient landslides, such as the Jiaju landslide, Niela landslide, Moluo landslide, and Jianshejie landslide (Figure 2a). Because of neotectonic movement, river erosion and extreme rainfall were influenced, various ancient landslides were prone to reactivation, and some were in deep creeping and large deformation states. Since 2006, the Jiaju landslide has developed evident traces of deformation and damage, and the leading-edge instability has occupied the Dajin River. This landslide is still creepy, the houses on the body are cracked, and the road is severely squeezed and deformed.

## 3. Methods and Data

This study used various methods to investigate and research the Jiaju ancient landslide, such as field surveys, unmanned aerial vehicles (UAVs), remote sensing interpretation, and synthetic aperture radar interferometry.

### 3.1. Geological Survey

The field surveys for geological hazards judged and analyzed the geomorphological settings and deformation zone based on vegetation characteristics, damage to buildings, the shape of the landslide body, and the rock mass structure. In addition, this study used a geological map based on the relevant requirements of the "Specification of comprehensive survey for landslide, collapse and debris flow (1:50,000)" (DZ/T 0261-2014) to carry out the field surveys of the development characteristics of landslides. UAV was used to investigate and map the potential sliding and assess hazards: a high-resolution (10~20 cm) digital image and a digital surface model (10~20 cm) were obtained by the DJI Phantom 4 UAVs, fabricated by the Da-Jiang Go. China, with a maximum flying flight of 500 m, and the image sensor is 12.4 million pixels.

### 3.2. SAR Data

This study collected 96 ascending Sentinel-1A single look complex (SLC) SAR data acquired in IW mode and VV+VH polarization. SAR data cover 39 months, from 24 June 2018 to 31 August 2021. The SAR images were cropped as a 30 × 30 km area centered on the Jiaju ancient landslide (Figure 1). The average incidence angle in the study area is 43.9° from the vertical direction (Table 1). The pixel spacing is 2.32 m in the range direction and 13.93 m in the azimuth direction.

**Table 1.** The basic parameters of SAR image data.

| SAR Sensor | Parameters |
| --- | --- |
| Direction | Ascending |
| Path | 26 |
| Frame | 93 |
| Wave band | C |
| Radar wavelength/(cm) | 5.6 |
| Incident angle/(°) | 43.90 |
| Time interval/(day) | 12 |
| Image time | 24 June 2018 to 31 August 2021 |
| Quantity of SAR images | 96 |

### 3.3. SBAS-InSAR Method

In this study, the SBAS-InSAR method is used to analyze the deformation of the Jiaju ancient landslide, the basic principle is as follows (Figure 3).

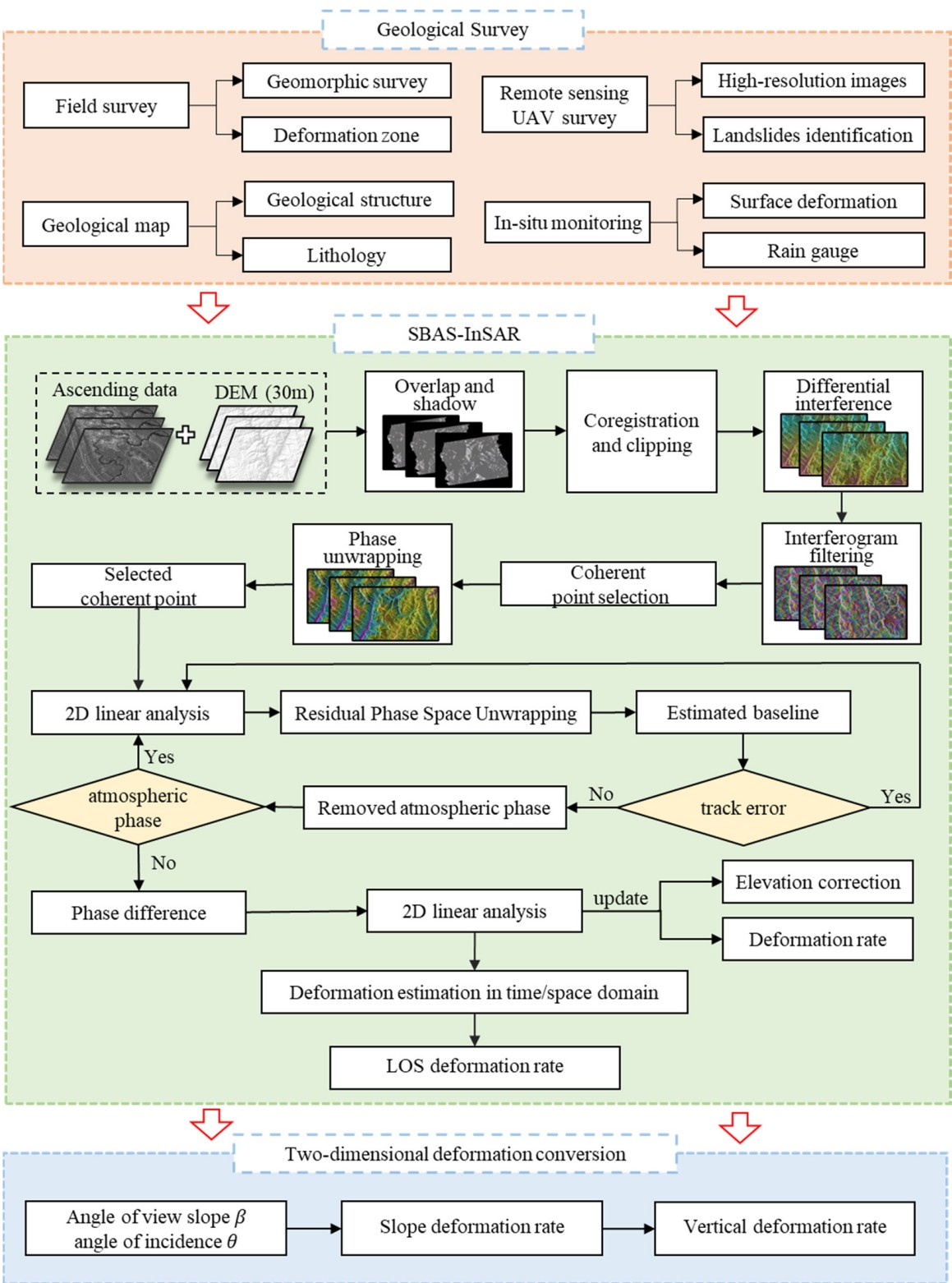

**Figure 3.** Research methodology map.

On the basis of acquiring N + 1 SAR images in the study area, the appropriate primary and secondary images are determined for registration, and *M* interferograms are obtained. Suppose *t* is the acquisition time of the images (sorted by time). After removing the flat ground effect and terrain phase, the interference phase of the image is generated by the pixel

point $(x, r)$ ($x$ is the azimuth, $r$ is the range) at $t_A$ and $t_B$ moments, and then Equation (1) is determined:

$$\delta\varphi_{\mathrm{j}}(x, r) = \varphi(t_B, x, r) - \varphi(t_A, x, r)$$
$$\approx \tfrac{4\pi}{\lambda}[\Delta d_{disp}(t_{B-A}, x, r)] + \tfrac{4\pi}{\lambda} \cdot \tfrac{B_{\perp j}\Delta h}{r\sin\theta} + \tfrac{4\pi}{\lambda}[\Delta d_{atm}(t_{B-A}, x, r)] + \Delta n_{\mathrm{j}} \tag{1}$$

In Equation (1), $\delta\varphi_{\mathrm{j}}$ $(x, r)$ is the interference phase at the pixel; $\varphi$ $(t_B, x, r)$ and $\varphi$ $(t_A, x, r)$ are the phases of the pixels at times $t_A$ and $t_B$, respectively; $j$ is the number of differential interferograms; $\lambda$ is the center wavelength; $\Delta d_{disp}$ is the deformation phase difference; $\Delta h$ is the terrain phase difference; and $\Delta d_{atm}$ is the atmospheric phase error. After removing the terrain phase error and the atmospheric phase error, an equation group can be obtained from Equation (1), in which there are $M$ equations and $N$ unknowns, $\varphi$ $(t_i, x, r)$ ($i = 1, \ldots, M$), and then Equation (2) can be constructed:

$$A\varphi = \delta\varphi \tag{2}$$

In Equation (3), $B$ represents the $M \times N$ matrix. If the SAR data are divided into several independent small baseline sets, it causes the rank deficit of matrix $B$ and leads to infinite solutions of matrix Equation (3). However, using the SVD method in the SBAS processing procedures to perform a pseudoinverse operation on matrix $B$, the least squares solution of Equation (3) can be obtained [30].

$$Bv = \delta\varphi \tag{3}$$

Time-series analysis was performed using the SBAS-InSAR method implemented in GAMMA software [25].

First, the $10 \times 2$ multilook was used for Sentinel-1 data to determine the resolution ratio proportionate relationship between the azimuth and range direction of a single look complex. The spatial and temporal baselines are 150 m and 24 days, respectively, to ensure the coherence of the interferogram. Combined with 30 m resolution shuttle radar topography mission (SRTM), digital elevation model DEM data from the NASA website (https://lpdaac.usgs.gov, accessed on 25 September 2021) were used to remove terrain phase mission. The minimum cost flow method is used for phase unwrapping to ensure the accuracy of the obtained unwrapping phase. A coherence higher than 0.5 is used as the masking threshold for unwrapping. Here, the precise orbital parameter information of Sentinel-1 data was used to remove the orbital errors.

Second, the trend errors are filtered out utilizing high and low pass filtering, such as residual orbital and atmospheric phases. According to the interferogram and phase average rate, and in the filtering process, the atmospheric phase is estimated in the stable region to reduce the phase loss of the deformation zone. The point density obtained in this study was 954 points/km$^2$.

Third, several small baselines were jointly solved using the singular value decomposition (SVD) method, and the phase rates obtained each time were integrated into the time domain. Then, the LOS deformation time was obtained for the Jiaju ancient landslide from June 2018 to August 2021 (Figure 3 and Table 2).

**Table 2.** The parameters of SBAS-InSAR progressing.

| Progressing | Parameters |
| --- | --- |
| multi-look | $10 \times 2$ |
| temporal baselines | 150 m |
| spatial baselines | 24 days |
| DEM resolution | 30 m |
| masking threshold | 0.5 |
| interferograms | 96 |

Considering the geometric relationship between the radar LOS direction, slope direction, and vertical direction, it is assumed that the landslide movement occurs along the direction specified by the unit vector û. Through the two-dimensional (2D) deformation conversion, Equations (4)–(7) [44] are used to transform the LOS ($V_{LOS}$) direction movement velocity into the slope direction ($V_{Slope}$) and the vertical direction ($V_{Vertical}$).

$$\hat{u} = \begin{bmatrix} -\sin\alpha\cos\varphi \\ -\cos\alpha\cos\varphi \\ \sin\varphi \end{bmatrix} \tag{4}$$

$$\cos\beta = \begin{bmatrix} (-\sin\alpha\cos\varphi)(-\sin\theta\cos\alpha_s)+ \\ (-\cos\alpha\cos\varphi)(\sin\theta\sin\alpha_s)+ \\ (\sin\varphi\cos\theta) \end{bmatrix} \tag{5}$$

$$V_{Slope} = V_{LOS}/\cos\beta \tag{6}$$

$$V_{Vertical} = \frac{V_{LOS} + V_{Slope}\sin\theta\cos\left[\delta - \left(\alpha_S - \frac{3}{2}\pi\right)\right]}{\cos\theta} \tag{7}$$

where $V_{Slpoe}$ is the slope deformation rate, $V_{LOS}$ is the radar *LOS* deformation rate, $V_{Vertical}$ is the vertical deformation rate, $\alpha_S$ is the angle between the azimuth and the actual north direction, $\alpha_S - 3/2\pi$ is the azimuth to *LOS*, $\delta$ is the azimuth angle of the landslide, $\alpha$ is the aspect, $\beta$ is the angle between *LOS* and slope, and $\theta$ is the incident angle.

For the abnormal phenomenon of absolute value in the conversion process from $V_{LOS}$ to $V_{Slope}$, Herrera et al. [45] proposed cos$\beta$ = ±0.3 as a fixed threshold, that is, $V_{Slope}$ cannot be greater than 3.33 times $V_{LOS}$, and set it when −0.3 < cos$\beta$ < 0, cos$\beta$ = −0.3; when 0 < cos$\beta$ < 0.3, cos$\beta$ = 0.3 [46].

### 3.4. SAR Data Suitability Analysis

In mountainous areas with large terrain fluctuations, the orientation and slope will become the decisive factors of the landslide, which could determine whether geometric distortion will occur. The Jiaju ancient landslide faced away from the satellite and had a gentle slope (Figure 4a). Normal SAR imaging is a relatively optimal observation. At this time, the SAR imaging resolution is relatively high, the geometric distortion is low, and the local incident angle $\theta$ is larger than the satellite incident angle (43.9°) and less than 90°, which is suitable for SAR observations in this orbit. This study calculated the distribution of the Sentinel-1A ascending data covering the Jiaju ancient landslide, including overlapping areas, shadow areas, and suitable areas (Figure 4b). In ascending data, recognition of the suitable area was 97%, the overlapped area was 1%, and the shadow area was 2%, which is beneficial to identify the Jiaju ancient landslide deformation area.

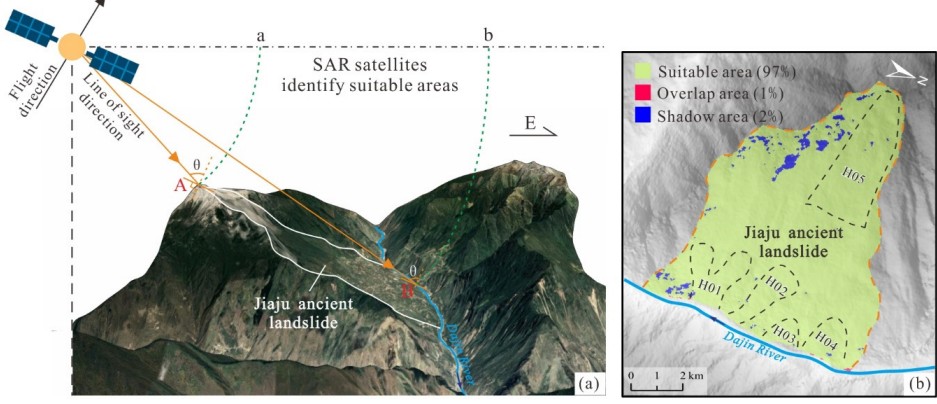

**Figure 4.** Suitability analysis of deformation monitoring in Sentinel-1A ascending data. (**a**) the principle of SAR side-view imaging in the Jiaju ancient landslide; (**b**) geometric distortions of Sentinel-1 ascending image along the Jiaju ancient landslide.

## 4. Results Analysis

### 4.1. Engineering Geological Characteristics

4.1.1. Spatial Development Characteristics

The Jiaju ancient landslide presents an armchair-shaped plane with an elevation varying from 1902~4459 m, and the height difference is approx. 2500 m. The average slope of the landslide is approx. 20°, and its direction is approx. 93°. The Jiaju ancient landslide is approximately 8 km in length and 6 km in width, and the area is approximately $3200 \times 10^4$ m$^2$, by drilling in this area, the average sliding depth is revealed to be 22 m [36]. This ancient landslide is assessed to be approx. $7.04 \times 10^8$ m$^3$, which is a giant ancient landslide. There is a stepped terrace landform coexisting with wide and gentle slopes and steep sills. The gully is relatively developed on the landslide body, with a 5~12 m depth. There are more than ten perennial flowing streams in the middle of the slope.

Based on the field surveys and remote sensing interpretation, the landslide could be divided into three parts: formation area (I), sliding area (II), and accumulation area (III) (Figure 2b). The elevation of sliding area (II) is 2900~3600 m, and the elevation of front fringe accumulation area (III) is 1900~2900 m. The Jiaju ancient landslide is composed of five secondary landslides, the Jiaju landslide (H01), Niexiaping landslide (H02), Xiaobawang landslide (H03), Niela landslide (H04), and Mt.-peak landslide (H05) (Figure 2b).

4.1.2. Development Characteristics of Five Secondary Landslides

(1)　Jiaju landslide (H01)

The Jiaju landslide (H01) presents an "M" shape in the plane (Figure 2c), and it could be divided into northern and southern parts. The landslide is approx. 1100~1200 m length and 800~1000 m width. The sliding area is approx. 1.2 km$^2$, based on field surveys and drilling results, the mean thickness of the landslide is approx. 22 m and the volume is approx. $2.6 \times 10^7$ m$^3$ [37]. The bedrock is exposed on both sides and the central ridge. The rock occurrence is mainly NE and NW, the dip angle is 35°~60°, and the thickness of the strongly weathered rock is 8~13 m. The main movement of the deformation area is concentrated in the northern part of the landslide and is mainly manifested as multilevel and multistage deformation inside the landslide.

(2)　Niexiaping landslide (H02)

The Niexiaping landslide is located in the northern part of the Jiaju ancient landslide, with a long tongue-shaped landform on the plane. The elevation of the landslide is approx. 1890~2530 m, the difference in elevation is approx. 650 m, and the longitudinal slope of the landslide is approx. 20°~30°, the primary sliding orientation is 100°, and the east-to-west length is approx. 2200 m, and the north-to-south width is approx. 1000 m, the area is approx. $2.1 \times 10^6$ m$^2$, and the estimated volume is approx. $1.1~1.5 \times 10^8$ m$^3$.

(3)　Xiaobawang landslide (H03)

The Xiaobawang landslide is located between the Niexiaping landslide (H02) and the Niela landslide (H04), which is 730 m in length and 820 m in width, with armchair-shaped geomorphology. The aspect is 103°, and the slope is approximately 15°~25°. The elevation at the front is 1906 m, the height of the posterior border of the landslide is approximately 2161 m, and the elevation drop is 260 m.

(4)　Niela landslide (H04)

The Niela landslide is on the north side of the Jiaju ancient landslide. It is typical armchair-shaped, 1100 m long, and 1200 m wide, and the area is approximately $1.3 \times 10^6$ m$^2$. The estimated volume is approximately $2.6~3.0 \times 10^7$ m$^3$. The field surveys found that the road subsided to the outside, at the front edge of the landslide, the road subsided 20 m (Figure 2f), and the Dajin River eroded the front edge of the landslide.

(5)    Mt.-peak landslide (H05)

The deformation region of the Mt.-peak landslide is at the western top of the Jiaju ancient landslide. It has a long tongue-shaped geomorphology. The landslide elevation is between 2900 and 4400 m, and the elevation drop is 1500 m, with a 4200 m length and 1500 m width. The landslide area is approximately $6.3 \times 10^6$ m$^2$, and it is estimated that the landslide volume is approximately 9~11 $\times 10^8$ m$^3$, which is a large-scale landslide. Xu et al. [43], through a LiDAR image, revealed an apparent phenomenon: the Mt.-peak landslide was broken at the rear edge and had a secondary collapse phenomenon, and the leading edge bulged and uplifted (Figure 2e).

### 4.2. Deformation Characteristics of the Jiaju Ancient Landslide

4.2.1. The Jiaju Ancient Landslide Deformation in the LOS

According to the LOS direction deformation results of the Jiaju ancient landslide (Figure 5), the maximum deformation value is −179 mm/a. According to the spatial differences in the overall deformation movement characteristics of the Jiaju ancient landslide, the deformation region is mainly distributed on the north side of the Jiaju landslide (H01), the south side of the Niela landslide (H04) and the Mt.-peak landslide (H05). The Niexiaping landslide (H02) and the Xiaobawang landslide (H03) only have slight deformation in the middle and the whole is relatively stable. Among the three secondary deformation rates of H01, H04, and H05, the deformation rate of H04 is up to −170 mm/a (Figure 5), which is more than three times larger than that of H01. The main factors that induce landslide deformation are river erosion, and human engineering activities on the slope, which indicate that the area is a hazard prevention area requiring high attention.

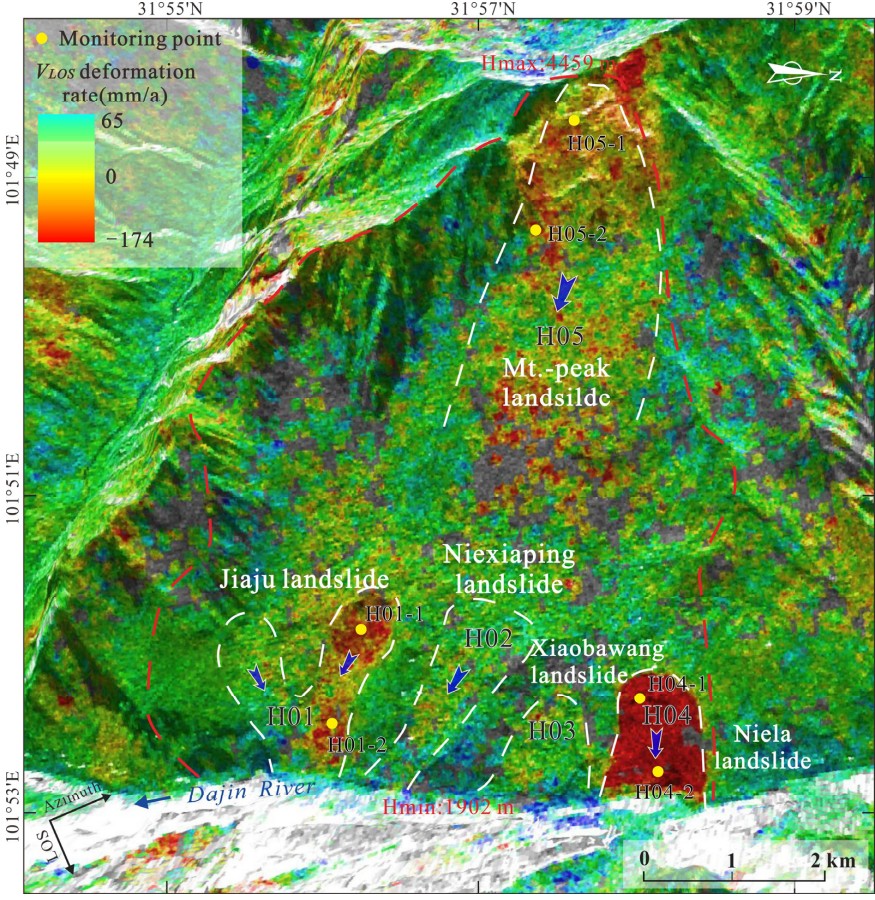

**Figure 5.** Deformation rate distribution map of the Jiaju ancient landslide in the LOS direction.

### 4.2.2. Two-Dimensional (2D) Deformation Rate Characteristics of the Jiaju Ancient Landslide

To compare the measurement results of the ground-surface systems with the InSAR measurements, the velocity along the LOS ($V_{LOS}$) was converted to the slope ($V_{Slope}$) (Figure 6a) and vertical ($V_{Vertical}$) (Figure 6b). The H01~H05 landslides showed negative values along the slope direction ($V_{Slope}$) (Figure 6a and Table 3), indicating that the secondary landslides were moving along the slope direction, and the maximum deformation velocity was −211 mm/a in the slope direction. The deformation rate displayed from orange to red ($V_{Slope} < -40$ mm/a) among the north side of the Jiaju landslide (H01), the whole of the Niela landslide (H04) and the Mt.-peak landslide (H05), indicating that the secondary landslide has a significant deformation velocity along the slope direction (Figure 6a). The north side of the Jiaju landslide (H01), the Niela landslide (H04), and the Mt.-peak landslide (H05) tend to move in the vertical direction and the maximum velocity reaches −67 mm/a (Figure 6b). The Niexiaping landslide and the Xiaobawang landslide changed from blue to green (−10~10 mm/a) in the vertical direction, and the deformation value was small and not noticeable.

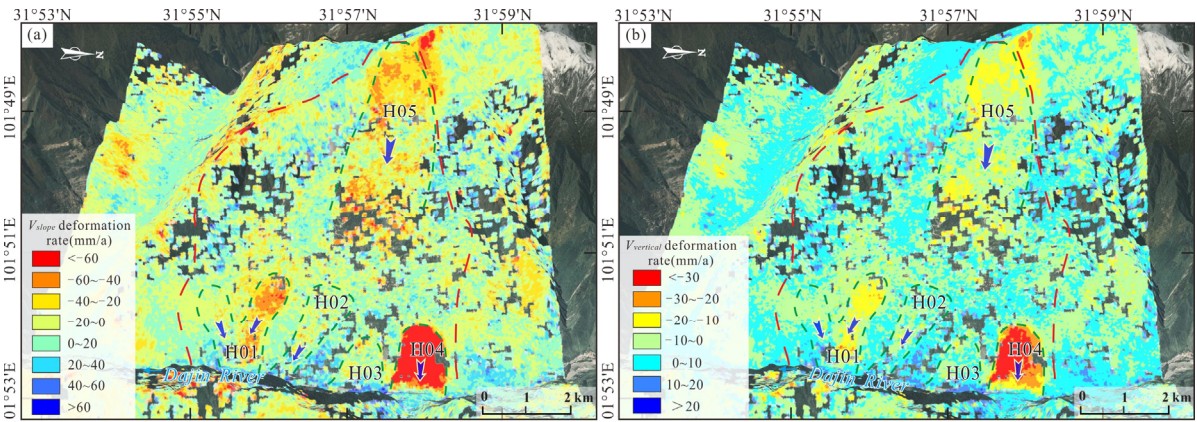

**Figure 6.** Two-dimensional deformation rate of the Jiaju ancient landslide; (**a**) slope direction; (**b**) vertical direction.

**Table 3.** The monitoring and identification results of the Jiaju ancient landslide based on InSAR.

| Name | Aspect (°) | Slope (°) | Area (km²) | Maximum LOS Rate (mm/a) | Maximum Slope Rate (mm/a) | Maximum Vertical Rate (mm/a) |
|---|---|---|---|---|---|---|
| Jiaju landslide (H01-N) | 112 | 20~35 | 0.6 | −92 | −108 | −35 |
| Jiaju landslide (H01-S) | 67 | 20~35 | 0.8 | −42 | −50 | −16 |
| Niexiaping landslide (H02) | 120 | 20~35 | 2.1 | −44 | −52 | −17 |
| Xiaobawang landslide (H03) | 103 | 15~25 | 1.1 | −32 | −66 | −22 |
| Niela landslide (H04) | 87 | 15~30 | 1.3 | −170 | −211 | −180 |
| Mt.-peak landslide (H05) | 98 | 20~40 | 6.3 | −87 | −103 | −33 |

### 4.2.3. Typical Section Deformation Characteristics of the Jiaju Ancient Landslide

The A-A′ line along the sliding direction profile (Figure 2b) is from the Mt.-peak landslide to the Niela landslide. We plotted the profile distribution of the deformation rate along section A-A′ in the three directions of $V_{LOS}$, $V_{Slope}$, and $V_{Vertical}$ (Figure 7). The deformation rate shows strong fluctuations along the A-A′ profile. The deformation values in the three directions are concentrated on the Niela landslide and the front edge of the Jiaju ancient landslide, and the maximum value of $V_{LOS}$ is over −140 mm/a (Figure 5). The Mt.-peak landslide at the trailing edge of the mountain also has deformation, and the value in the middle of the slope fluctuates up and down. The distribution characteristics of the

deformation values on the profile are different and are seriously affected by the different positions of the landslide and the local topography of the surface.

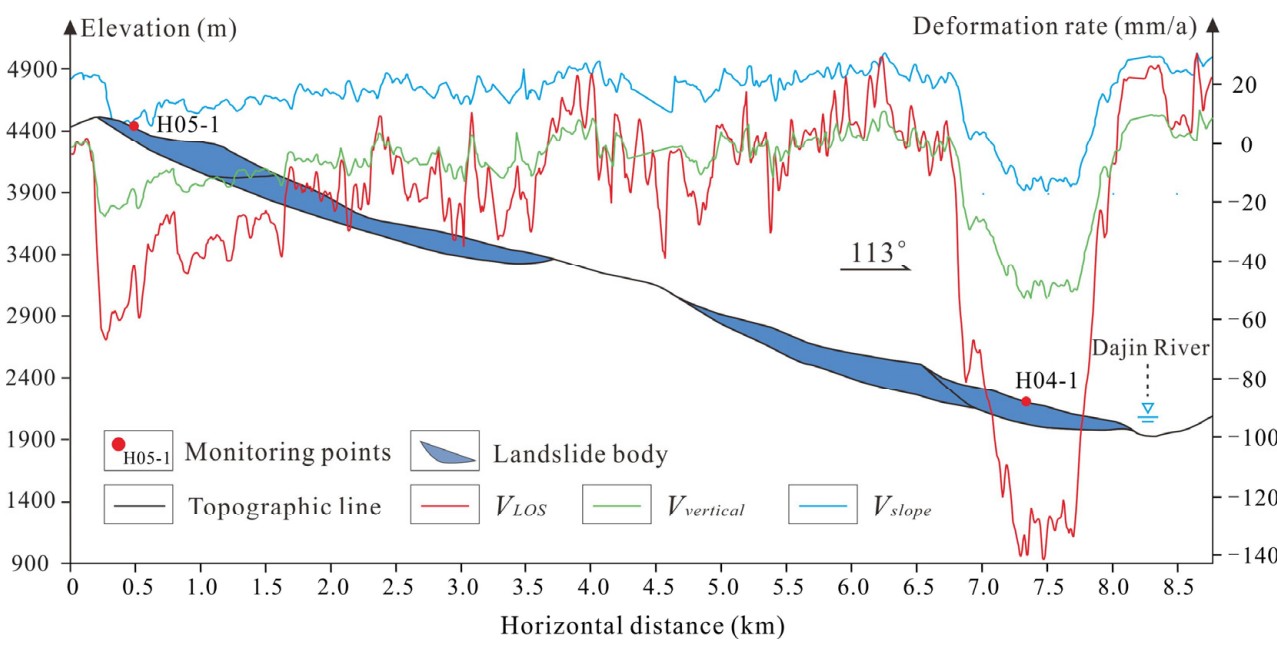

**Figure 7.** Deformation rate along the A-A′ profile (A-A′ profile in Figure 2b).

### 4.3. Deformation Characteristics of the Jiaju Landslide (H01)

The InSAR deformation of the Jiaju landslide (Figure 8a) shows a more significant difference between H01-N and H01-S. Green indicates that this area is relatively stable, and red to yellow indicates landslide movement away from the LOS direction. The Jiaju landslide could have diverged into three secondary sliding zones: J01, J02, and J03. The J01 zone is on the south flank, and the slope is relatively stable, with a maximum deformation rate of −30 mm/a, in comparison, the J02 and J03 zones are −72 mm/a and −92 mm/a, respectively. The field surveys found that the road at the trailing edge of J02 collapsed, three cracks appeared in the retaining wall, and the front edge was precarious (Figure 8b,c).

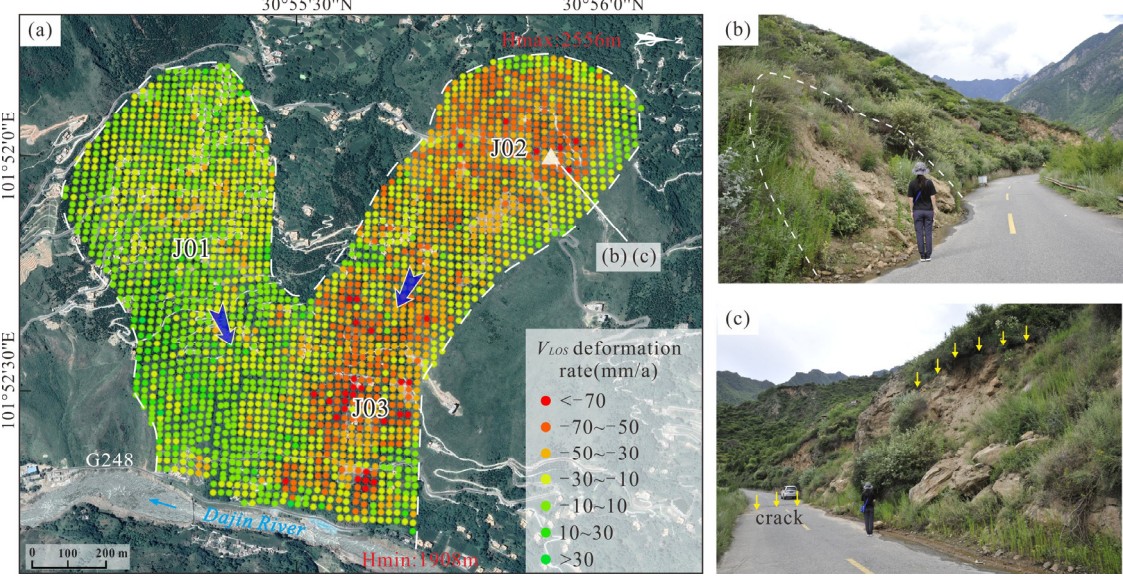

**Figure 8.** Deformation of the Jiaju landslide (H01); (**a**) InSAR deformation map of the Jiaju landslide; (**b**) the rear edge collapse (mirror to NE); (**c**) road cracks and rockfall (mirror to N).

The deformation trend of H01-N in this study is up to −35 mm/a, which is smaller than that of Yin et al. [36], who obtained 43.8 mm/a, possibly because there were some geohazard management and prevention projects applied in the H01 area, which slowed the deformation trend. The deformation velocity of the Jiaju landslide is greater in the north than in the south, and the front edge is greater than the rear edge, which is basically consistent with previous studies [36,38,47].

### 4.4. Deformation Characteristics of the Niela Landslide (H04)

According to the deformation characteristics of the Jiaju ancient landslide (Figure 5), the Niela landslide is in a creep state. The orange to red area ($V_{LOS} < -60$ mm/a) indicates this area with a significant deformation velocity of the landslide (Figure 9a). The deformation is more substantial on the south side than on the north side and the rear edge than on the front edge, and the maximum deformation rate is −212 mm/a. Therefore, it is hypothesized that the Niela landslide presents push-type creep. Furthermore, field surveys and UAV images showed that the front edge of the landslide is undergoing large displacement and deformation by the Dajin River erosion, and there are nine potentially unstable collapsed bodies ①~⑨ located at the toe of the slope (Figure 9c). The most significant collapse height is approx. 100 m.

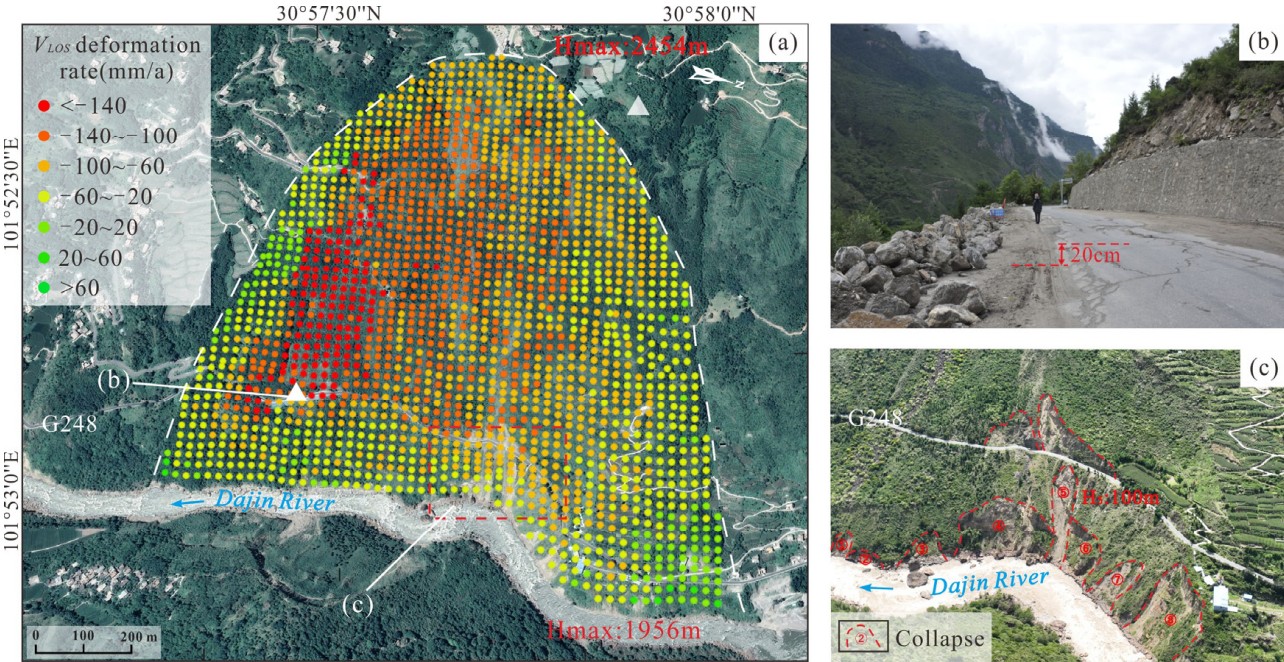

**Figure 9.** Deformation of the Niela landslide (H04); (**a**) InSAR deformation map of the Niela landslide; (**b**) road subsidence phenomenon (mirror to SE); (**c**) nine collapses at the front edge (based on UAV).

### 4.5. Deformation Characteristics of the Mt.-Peak Landslide (H05)

It can be seen from the LOS results that the whole deformation results of the Mt.-peak landslide are red to green (Figure 5), the deformation velocity is negative, and the maximum velocity exceeds −80 mm/a. It also shows a negative value in the slope direction (Figure 6a), indicating that the landslide has a downward sliding trend along the slope and the maximum slope deformation rate $V_{Slope}$ up to −103 mm/a. The deformation in the H05 area is mainly because of the high altitude and the decrease in the mechanical strength of the rock mass caused by freeze-thaw effects.

## 5. Discussion

The deformation mechanisms and creep trends in ancient landslide studies have always been the focus of engineering geology and geological hazards [20]. This study studied three parts of the Jiaju ancient landslide: the deformation trend, the impact of extreme rainfall, and the landslide stability.

### 5.1. Deformation Trend of the Jiaju Ancient Landslide

#### 5.1.1. Landslide Creep Zone

According to the analysis of InSAR deformation, Herrera et al. [45] considered that the landslide stability is poor when the absolute value of the deformation in the LOS velocity of the L-band $V_{LOS-L} \geq 21$ mm/a, the C-band $V_{LOS-C} \geq 14$ mm/a, and the X-band $V_{LOS-X} \geq 6$ mm/a. According to the whole deformation of the Jiaju ancient landslide, referring to the deformation classification results of Feng et al. [46] for creeping landslides, combined with the field surveys in this area, the deformation level of the Jiaju ancient landslide is divided into four levels: extremely strong deformation ($-179$ mm/a $\leq V_{LOS} < -37$ mm/a), strong deformation ($-37$ mm/a $\leq V_{LOS} < -18$ mm/a), medium deformation ($-18$ mm/a $\leq V_{LOS} < 17$ mm/a), and low deformation ($17$ mm/a $\leq V_{LOS} < 61$ mm/a).

#### 5.1.2. Deformation Trend of the Jiaju Landslide (H01)

Deng et al. [48] set up eleven GPS monitoring points in the northern region of the Jiaju landslide. The monitoring value demonstrated that it is the northern portion of the landslide deformation, the horizontal displacement speed in the north is up to 53~200 mm/a, and the vertical displacement speed is up to $-3\sim-21$ mm/a. Since 2006, the Jiaju landslide has exhibited evident traces of deformation, and the toe of the landslide has slipped to the Dajin River. It is still in a creepy state. Through the cumulative deformation map of the Jiaju ancient landslide at different periods (interval 120 days) (Figure 10), there is a trend of deformation accumulation in the northern portion of the Jiaju landslide, the maximum cumulative deformation reaches 250 mm, the south side is relatively stable, and the over three years accumulative deformation is approximately 50 mm. Based on the warning systems calculated the warning parameters in the Jiaju landslide, the ratio of displacement change to unitary measurement time ($\Delta v$) is approximately 0.15 mm/d, and the tangential is approximately 32°. It is considered that the landslide is in creep deformation.

#### 5.1.3. Deformation Trend of the Niela Landslide (H04)

According to the time series accumulative deformation (Figure 10), the Niela landslide has noticeable deformation accumulation. The deformation on the south side of the landslide is stronger than that on the north side, and the north part is stable and starts to creep. The cumulative deformation curve grows exponentially (Figure 11), and the maximal accumulative deformation reaches 861 mm. The calculated ratio of displacement change to unitary measurement time ($\Delta v$) in the Niela landslide is approximately 0.8 mm/d, and the tangential is approximately 55°. It is considered that the landslide is in creep deformation.

### 5.2. Impact of Extreme Rainfall on the Deformation

The Jiaju ancient landslide is in a specific Tibetan Plateau monsoon climate region [49]. The vertical temperature changes significantly, and the annual average rainfall is 600 mm. For example, heavy rainfall triggered a debris flow in Meilonggou, causing the front fringe of the Aniangzhai ancient landslide to collapse and dam the Xiaojin River. This study concluded the connection between the daily precipitation and the accumulative deformation of the Jiaju ancient landslide from June 2018 to August 2021 (Figure 11). The deformation rates of H01-1 and H01-2 are the same, and after heavy rainfall, the cumulative deformation of H01-1 shows a steady increase, and the cumulative deformation of H01-2 increases greatly (Figure 11). When the front edge velocity is more significant than the rear edge velocity, the front edge is eroded by the Dajin River, and the large strain concentration at the toe of the slope, may lead to progressive traction deformation from front to back in

the northern part of the Jiaju landslide. At the monitoring points (H04-1 and H04-2), in June 2020, due to heavy rainfall, the cumulative displacement of the landslide increased significantly, and the deformation rate was accelerated. The rainfall significantly promoted the surface displacement of the landslide. The weakening of the soil may reduce the stability of the landslide. The Dajin River perennially erodes the front edge of the landslide. Heavy rainfall will lead to accelerated river erosion, resulting in severe and continuous landslide deformation [49].

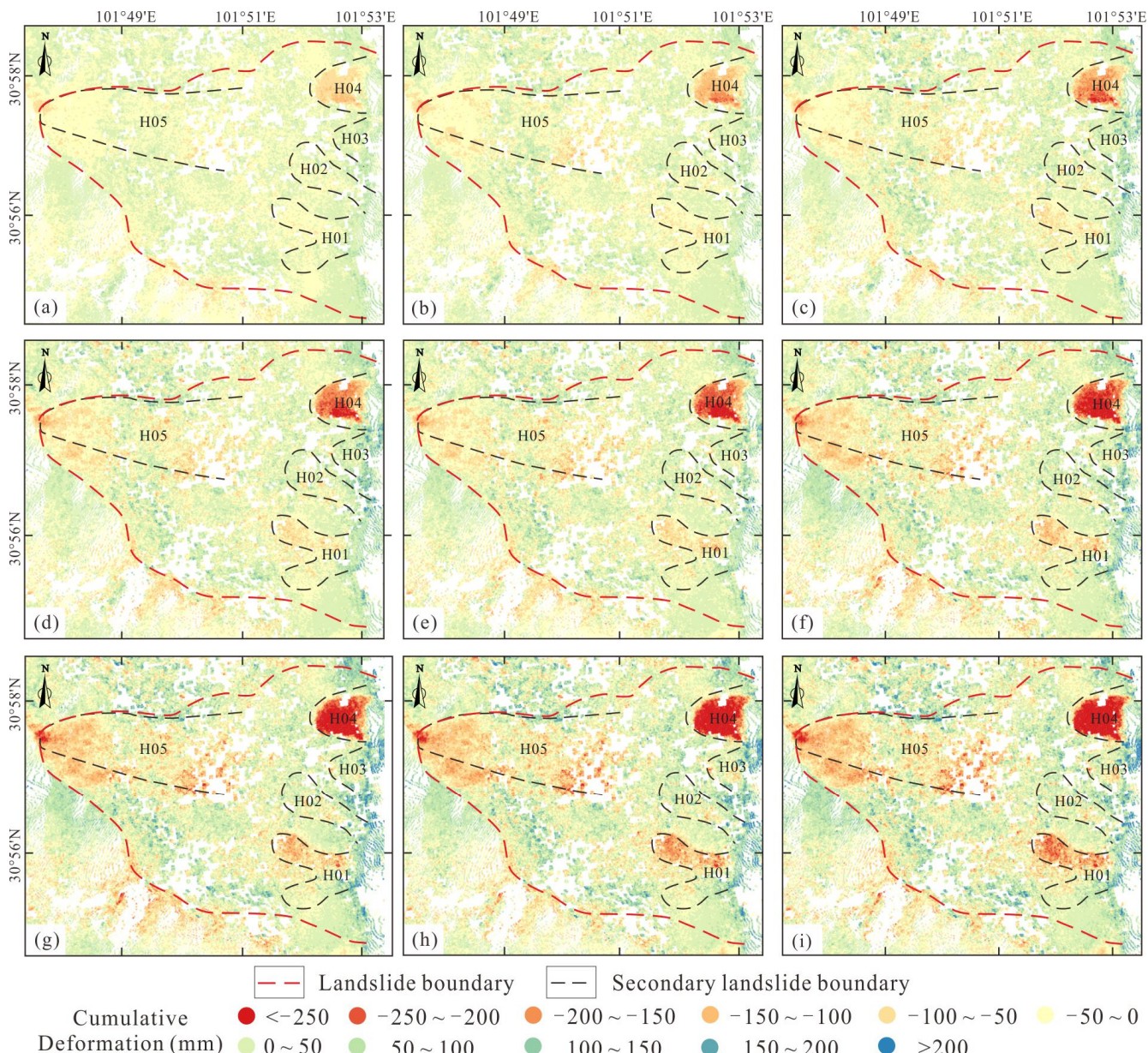

**Figure 10.** Cumulative deformation at different times calculated by InSAR (LOS); (**a**) 19 February 2019 (240 days); (**b**) 19 June 2019 (360 days); (**c**) 17 October 2019 (480 days); (**d**) 14 February 2020 (600 days); (**e**) 13 June 2020 (720 days); (**f**) 23 October 2020 (852 days); (**g**) 20 February 2021 (972 days); (**h**) 20 June 2021 (1092 days); (**i**) 31 August 2021 (1164 days).

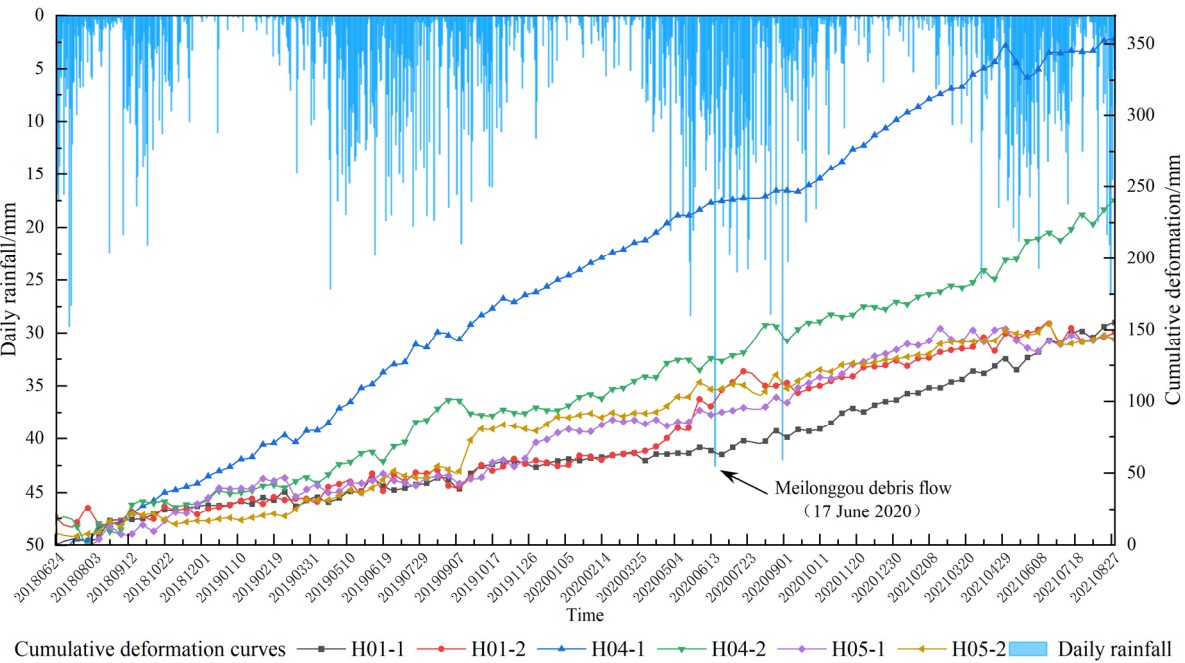

**Figure 11.** Daily precipitation and accumulative deformation curve of the Jiaju ancient landslide.

*5.3. Stability Trend Analysis of the Jiaju Ancient Landslide*

The formation and evolution of landslides could be revealed through long-term In-SAR data deformation monitoring. However, there is still considerable controversy about when a large-scale landslide overall slip will occur after long-term creep of a landslide. Important factors, such as the terrain and geomorphology of the landslide area, the lithology of the strata, the instability mechanism, the accuracy of the monitoring results, and the deformation rate of the monitored objects influence the critical sliding thresholds of stable and unstable landslides, the critical thresholds such as surface displacement, speed, and acceleration.

Intrieri et al. [50] and Allasia et al. [51] proposed a threshold breakthrough model for landslide warning. Xu et al. [52] calculated the velocity of steady-state creep $v_0$ by averaging vast amounts of steady-state measured data, obtained a reliable baseline, calculated the displacement rate $\Delta v$, $\Delta v = v \div \Delta t$ (that is, the ratio of displacement change to unit time), used the formula $\alpha = \arctan(\Delta v / v_0)$ to calculate the tangential angle $\alpha$, and compared the values of $\Delta v$ and $\alpha$ with predefined thresholds to issue early warnings (Figure 12). Xie et al. [53] studied the reactivation characteristics of a large ancient landslide in Wenchuan through InSAR deformation monitoring and a real-time monitoring system of landslide site displacement. The long-term InSAR displacement and short-term real-time monitoring curves were compared with the early warning model curve, and it was considered that the sizeable ancient landslide only accelerated briefly during heavy rainfall, and the overall deformation rate was stable.

The five secondary landslides of the Jiaju ancient landslide deform at different rates. The cumulative deformation curve (Figure 11) is compared with the early warning model of landslide deformation monitoring (Figure 12). The Niela landslide (H04) will have a short acceleration during heavy rainfall, and the rainfall and the displacement change rate are positively correlated, $\Delta v = 0.8$, the tangential angle $\alpha$ is approximately $55°$, but the overall displacement change rate is constant, so it is speculated that the rear edge of the Niela landslide is in the initial acceleration of accelerative deformation and at the Warn warning level in the C-D section (Figure 12). The warning parameters in the Jiaju landslide (H01) are calculated as $\Delta v \approx 0$ and $\alpha \approx 32°$, which show that this landslide is in a condition of creep deformation and attention warning level, in the B-C sections (Figure 12). For the Mt.-peak landslide (H05), $\Delta v < 0$, $\alpha$ is an invalid value, and this landslide is in the initial deformation

and Safety warning level, in sections A-B (Figure 12). In addition, the Niexiaping landslide (H02) and Xiaobawang landslide (H03) are in the potential development stage and at the safety warning level, in the O-A sections (Figure 12).

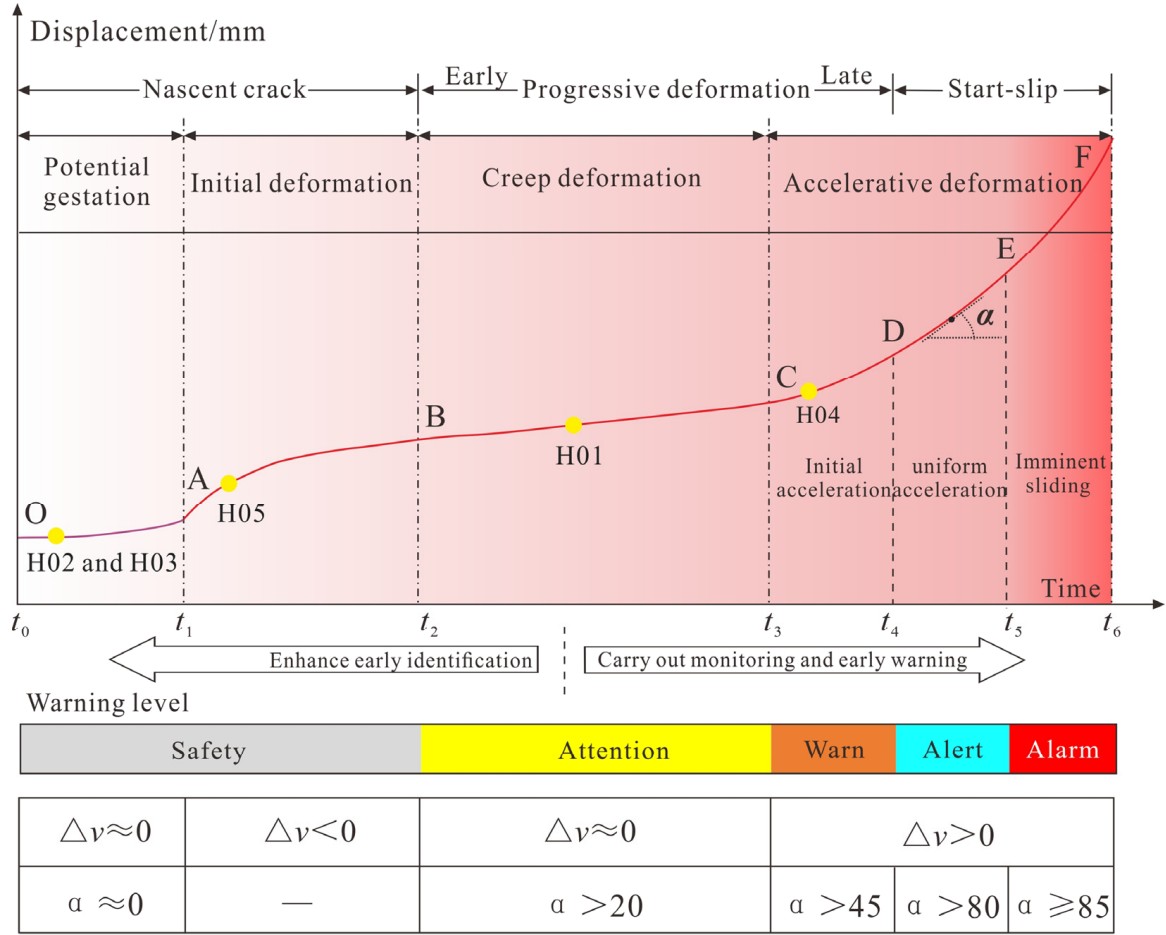

O-A:potential gestation  A-B:initial deformation  B-C:creep deformation  C-D:initial acceleration
D-E:uniform acceleration  E-F:imminent sliding  $\triangle v$:displacement growth rate  $\alpha$: tangential angle (°)

**Figure 12.** Deformation stage curve of a large deep-seated creeping landslide (modified after [43,54]).

## 6. Conclusions

Based on the SBAS-InSAR method, UAV, and field surveys, the Jiaju ancient landslide reactivation and deformation characteristics were studied in this study. The results and findings were concluded as follows.

First, the Jiaju ancient landslide is a complex giant ancient landslide and is mainly composed of the Jiaju landslide (H01), Niexiaping landslide (H02), Xiaobawang landslide (H03), Niela landslide (H04), Mt.-peak landslide (H05), and five secondary sliding bodies. From June 2018 to August 2021, the maximum velocities of the LOS, slope, and vertical directions of the Jiaju ancient landslide are −179 mm/a, −211 mm/a, and −67 mm/a, respectively.

Second, the northern area of the Jiaju landslide (H01), the southern area of the Niela landslide (H04) and the Mt.-peak landslide (H05) have large deformations. The deformation rate showed that the Jiaju landslide is dominated by traction deformation, and the Niela landslide is dominated by push deformation, and heavy rainfall will accelerate landslide deformation and trigger reactivation.

Third, the deformation of the Jiaju ancient landslide is divided into four levels: extremely strong deformation, strong deformation, medium deformation, and low deformation.

Fourth, the Niela landslide is in the initial acceleration state of accelerated deformation and the Warn warning level, the Jiaju landslide is in the creep deformation state and

Attention warning level, and the Mt.-peak landslide is in the initial deformation state and Safety warning level. Due to the complex geological structure and strong neotectonic movement, the secondary landslide creep-sliding rate of the Jiaju ancient landslide is accelerated under heavy rainfall and finally slides in part or as a whole, resulting in river blocking.

**Author Contributions:** Y.Y. participated in the field surveys, processed the InSAR data, and completed the manuscript writing. C.G. proposed the research concept, designed the framework, and revised the manuscript. C.L. participated in the field surveys and data analysis. H.Y. and Z.Q. participated in the field geology survey and revised the manuscript. All authors have read and agreed to the published version of the manuscript.

**Funding:** This research was funded by the National Natural Science Foundation of China (Nos. 41877279, 41731287, 41941017, 42007280), the China Geological Survey Project (Nos. DD20190319, DD20221816), and the Outstanding Young Scientific and Technological Talent Project of the Ministry of Natural Resources (No. 12110600000018003911).

**Data Availability Statement:** Thanks to the global availability of free and open Sentinel-1 SAR data with Europe Space Agency (ESA), the SAR data be accessible in https://sentinel.esa.int/web/sentinel/missions/sentinel-1, accessed on 18 September 2021.

**Acknowledgments:** We are very grateful to the reviewers who significantly contributed to the improvement of this paper. The authors would like to thank postgraduate students Ning Zhao, Yanan Zhang and Xiang Li for their help in the geology survey and further analysis.

**Conflicts of Interest:** The authors declare no conflict of interest.

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
