# Peer review of "The Creep-Sliding Deformation Mechanism of the Jiaju Ancient Landslide in the Upstream of Dadu River, Tibetan Plateau, China"

_remotesensing, doi:10.3390/rs15030592_

Round 1
Reviewer 1 Report
1. Please further clarify the differences of research contents and objectives between this manuscript and previous studies.
2. Further polish is needed for the English language.
3. The parameters in equation (1) and (2) were not described clearly.
4. Please add more information of SBAS-InSAR method. IPTA is different form SBAS. Please clarify the specific data processing method and steps used in this study.
5. The positive values in Figure 5 should be discussed more. Why?
6. In the result analysis part, the comparison between this study and the previous studies should be added.
7. How to validate the deformation results?
8. The slope and direction angle should be presented.
9. Please add some more descriptions about InSAR landslide monitoring in the introduction.
Reviewer 2 Report
The manuscript titled "The Creep-sliding Deformation Mechanism of the Jiaju Ancient Landslide in the Upstream of Dadu River, Tibetan Plateau, China" by Yan et al. presents a post-failure characterisation study of a landslide in China. The study utilises SAR satellite data from three years for this purpose. The authors have used the Advanced DInSAR (SBAS) technique on ascending the Sentinel-1 satellite. The Study is indeed very interesting to read. The authors have tried to present the study in a good way. The language used is easy to comprehend though it has a few mistakes, which I suggest can be done with proofreading by a third person. The discussion is very elaborately written. Still a few points I would try to raise/suggest:-
Specific Comments:
Why authors feel that “2D deformation transformation” should be a keyword?
Landslides are worldwide problems. The introduction should mention of landslide studies across the world, such as in Canada, Italy, Turkey, India, Pakistan, Spain, Argentina etc. Then proceed to China and Dadu river basin. In the Introduction, highlight the main contributions of the paper.
The authors can refer to other studies on Landslides of similar conditions in the Literature review (recent studies). This will surely strengthen the importance of the study, for, eg.
· Soltanieh, A., & Macciotta, R. (2022). Updated understanding of the Ripley landslide kinematics using satellite InSAR. Geosciences, 12(8), 298.
· Jia, H., Wang, Y., Ge, D., Deng, Y., & Wang, R. (2022). InSAR Study of Landslides: Early Detection, Three-Dimensional, and Long-Term Surface Displacement Estimation—A Case of Xiaojiang River Basin, China. Remote Sensing, 14(7), 1759.
· Mishra, V., & Jain, K. (2022). Satellite based assessment of artificial reservoir induced landslides in data scarce environment: A case study of Baglihar reservoir in India. Journal of Applied Geophysics, 205, 104754.
· Yao, J., Yao, X., & Liu, X. (2022). Landslide Detection and Mapping Based on SBAS-InSAR and PS-InSAR: A Case Study in Gongjue County, Tibet, China. Remote Sensing, 14(19), 4728.
the parameters used in the SBAS experiment should be given in a detailed table since the experiments have to be in a form that allows readers to repeat the experiments. What was the density of points obtained in the analysis? Authors are requested to comment upon it in the manuscript. Why not descending data used for the analysis? The authors must either justify only using the ascending track, for example, if the descending track is not available or should add the descending track data to the manuscript and attempt to fully resolve the movement of the landslide. This would make the result more convincing.
Use uniform units across the manuscript. Somewhere displacement rate uses mm/year and somewhere mm/a.
Under heading 3.1 “a high-resolution digital image and a digital surface model were obtained by UAV.” Is mentioned. What was the resolution of these products.
Figure 1 : Label names can be kept simply black. No need of white outline. Also, no need to mention path and data name in map. Mention it in the legend.
Figure 3: Don’t use bold letters. Mention which DEM you used in the figure.
Figure 5: There is something written in Chinese. Please translate it into English. Also, the verticle axis labels read like elevation per mm. Please denote it as ‘Elevation (mm)’.
Figure 11: the daily rainfall data can be depicted from upward. This will make the figure less clumsy. You can refer to Fig 11 of the following paper.
Mishra, V., & Jain, K. (2022). Satellite based assessment of artificial reservoir induced landslides in data scarce environment: A case study of Baglihar reservoir in India. Journal of Applied Geophysics, 205, 104754
In the sentence under heading 5.2, “This study statistics the connection between the daily precipitation and the accumulative deformation of the Jiaju ancient landslide from June 2018 to August 2021”, – the use of the word statistics is wrong. Rewrite the sentence.
Reviewer 3 Report
Excellent research and well written manuscript.
However, adding gird to figures 1, 5, 6, 8A, 9A, and 10 will improve location identification for readers not much familier with the area.
Figure 1: Please add grid to show lat/long information of the area. While cities can be shown as points, it is better if counties are shown as polygons. Similarly, study area may also be shown as polygon.
Figure 2: Legend is hardly readable. It is better to have north arrows pointing in the same direction for all components of the figure. No direction information is given in c, d, e, and f component of the figure.
Figure 4: It is better if the orientation of both a and b is same i.e., north arrow pointing up.
Figure 5: Please align LOS with LOS direction in figure 4 or align both with the flight path of Sentinel 1. Please also add lat/long grid.
Figure 6: Please add lat/long grid.
Figure 8: Please add lat/long grid to figure 8a. It is difficult to relate b and c with a. Please indicate on figure 8a the location a b and c.
Figure 9: Same comments as of figure 8.
Figure 10: Please add lat/long gird to all figures. North arrow alignment in all figures should be same. It is nice to have north arrow upward as it is in figure 10. Please make north arrow upward in all previous figures as well.
Please also add some description of the sensor mounted on UAV and the flight parameters.
Round 2
Reviewer 1 Report
The revised manuscript is suitable for publication.
Reviewer 2 Report
The manuscript can be pubished in its present form.